Sexing a sex-role-reversed species based on plumage: potential challenges in the red phalarope

Giroux Marie-Andrée 1 2 3 marie.a.giroux@gmail.com
Ditlecadet Delphine 4
Martin Luc J. 5
Lanctot Richard B. 6
Lecomte Nicolas 1 2 5 7
1 Canada Research Chair in Polar and Boreal Ecology, Université de Moncton , Moncton , Canada
2 Centre d’Études Nordiques, Université du Québec à Rimouski , Rimouski , Canada
3 Canada Research Chair on Northern Biodiversity, Université du Québec à Rimouski , Rimouski , Canada
4 Molecular Biology Unit, Fisheries and Oceans Canada , Moncton , Canada
5 Département de Biologie, Université de Moncton , Moncton , Canada
6 Migratory Bird Management, US Fish and Wildlife Service , Anchorage, AK , USA
7 Québec Center for Biodiversity Science, Université du Québec à Rimouski , Rimouski , Canada
Rebstock Ginger
Electronic publication date: 2016 May 2
Publication date: 2016
Volume: 4
Electronic Location ID: e1989
Received 2016 Feb 1; Accepted 2016 Apr 8
Copyright: ©2016 Giroux et al.
Copyright year: 2016
Copyright holder: Giroux et al.
License: This is an open access article distributed under the terms of the Creative Commons Attribution License, which permits unrestricted use, distribution, reproduction and adaptation in any medium and for any purpose provided that it is properly attributed. For attribution, the original author(s), title, publication source (PeerJ) and either DOI or URL of the article must be cited.
License URL: https://creativecommons.org/licenses/by/4.0/

Keywords: Charadriiformes, Phalaropus fulicarius, Shorebirds, Sexual dichromatism, Secondary sexual traits

Funding: The W. Garfield Weston Foundation (fellowship to MAG) Natural Science and Engineering Research Council of Canada Canadian Foundation for Innovation Polar Continental Shelf Project Canada Research Chair Program Government of Nunavut Indian and Northern Affairs Canada Igloolik Hunters and Trappers Organization US Fish and Wildlife Service Université de Moncton The funding sources for this study were: The W. Garfield Weston Foundation (fellowship to MAG), Natural Science and Engineering Research Council of Canada (EnviroNorth scholarship to MAG, Discovery grants to NL and LM), Canadian Foundation for Innovation (grants to NL and LM), Polar Continental Shelf Project (in-kind support to NL), Canada Research Chair Program to NL, Government of Nunavut (in-kind support), Indian and Northern Affairs Canada, Igloolik Hunters and Trappers Organization, US Fish and Wildlife Service, Université de Moncton. The funders had no role in study design, data collection and analysis, decision to publish, or preparation of the manuscript.

==============================
Sex-role reversal, in which males care for offspring, can occur when mate competition is stronger between females than males. Secondary sex traits and mate attracting displays in sex-role-reversed species are usually more pronounced in females than in males. The red phalarope (Phalaropus fulicarius) is a textbook example of a sex-role-reversed species. It is generally agreed that males are responsible for all incubation and parental care duties, whereas females typically desert males after having completed a clutch and may pair with new males to lay additional clutches. The breeding plumage of female red phalaropes is usually more brightly colored than male plumage, a reversed sexual dichromatism usually associated with sex-role reversal. Here, we confirm with PCR-based sexing that male red phalaropes can exhibit both the red body plumage typical of a female and the incubation behavior typical of a male. Our result, combined with previous observations of brightly colored red phalaropes incubating nests at the same arctic location (Igloolik Island, Nunavut, Canada), suggests that plumage dichromatism alone may not be sufficient to distinguish males from females in this breeding population of red phalaropes. This stresses the need for more systematic genetic sexing combined with standardized description of intersexual differences in red phalarope plumages. Determining whether such female-like plumage on males is a result of phenotypic plasticity or genetic variation could contribute to further understanding sex-role reversal strategies in the short Arctic summer.

Introduction

Sex-role reversal, in which males care for offspring, can occur when mate competition is stronger between females than males (Gwynne, 1991; Clutton-Brock & Vincent, 1991; Kvarnemo & Ahnesjo, 1996). Biases in the intensity of mating competition can result from differences in operational sex ratios (the ratio of males to females ready to mate), which can in turn be associated with biases in potential reproductive rates (Emlen & Oring, 1977; Kvarnemo & Ahnesjo, 1996). Theory predicts that, as a result of biases in the intensity of mating competition, secondary sex traits and mate attracting displays in sex-role-reversed species will be more pronounced in females than in males (Andersson, 1994; Eens & Pinxten, 2000; Trivers, 1985).

The red phalarope (Phalaropus fulicarius) is a textbook example of a sex-role-reversed species (Alcock, 2013). It is generally agreed that males are responsible for all incubation and parental care duties, whereas females typically desert males after having completed a clutch and may pair with new males to lay additional clutches (sequential polyandry; Dale et al., 1999; Schamel & Tracy, 1977). The mating system of the red phalarope has been described as female access polyandry, a system in which females do not defend resources, but rather limit access to males by converging at feeding areas to mate (Emlen & Oring, 1977). The breeding plumage of female red phalaropes is usually more brightly colored than male plumage (Tracy, Schamel & Dale, 2002; Fig. 1), a reversed sexual dichromatism usually associated with sex-role reversal (Heinsohn, Legge & Endler, 2005). It is also recognized that there is considerably more plumage variations among males, and that the most brightly colored males can approach female levels of coloration (Pyle, 2008; Tracy, Schamel & Dale, 2002). However, mottled crowns have been identified as the characteristic that was most diagnostic of males (Tracy, Schamel & Dale, 2002). Such overlap in the plumage of male and female red phalaropes (Tracy, Schamel & Dale, 2002) might explain why previous studies have reported incidental observations of red phalaropes showing typical female plumage either incubating eggs (3 out of 17 nests; Forbes et al., 1992) or brooding chicks (Sutton, 1932).

Figure 1 Comparison between the breeding plumage of three red phalaropes: (1) a typical male, (2) the ambiguous bird (brightly colored individual incubating), and (3) a typical female. All pictures were taken in Igloolik, Nunavut, Canada. Photos: N. Lecomte.

Here, we describe the observation of a red phalarope exhibiting both the red body plumage and the plain black crown of a female, but the incubation behavior typical of a male on Igloolik Island (Nunavut, Canada), during summer 2014. Our objective was to genetically sex this individual (hereafter referred to as the “ambiguous” individual) to determine whether it was a brightly colored male or a female. We determined the sex of the ambiguous bird by using a DNA marker universally used for sexing birds (Fridolfsson & Ellegren, 1999), comparing the band patterns of the ambiguous bird with those obtained with samples of red phalaropes sexed by dissection.

Methods

Study area

We conducted field work on Igloolik Island (Nunavut, Canada; 69°24′N, 81°32′W) between early June and early August in 2014 (Lecomte & Giroux, 2015). This island is located in northwest Foxe Basin next to the Melville Peninsula and south from the northern part of Baffin Island. The study area is located in a mosaic of wet (sedge/grass moss wetland), mesic (non-tussock sedge, dwarf-shrub, moss tundra), and dry (prostrate dwarf-shrub, herb tundra) habitat patches interspersed by ponds and lakes. We identified habitats as per the Circumpolar Arctic Vegetation map (CAVM-Team, 2003).

Nest monitoring

We located red phalarope nests by following birds on incubation recesses back to their nests or by flushing nests when walking or dragging a 30-m rope (9-mm-diameter). We searched for nests intensively within a 36-ha nest plot and a 24-ha nest plot, and also recorded the presence of nests found opportunistically outside of the nest plots. We recorded the location of each nest using a Global Positioning System (Garmin eTrex), and placed three nest markers at 1-m, 5-m and 10-m north of the nest to allow nest relocation. We monitored nests according to a 5-day visitation schedule.

Capture

We captured the ambiguous individual using a bownet placed on its nest on 16 July 2014 (1 day before hatching). We marked the bird with a metal band, a unique individual combination of three colored darvic bands, and a unique site-specific combination of two colored bands. We measured and recorded its bill length (exposed culmen) using a caliper (±0.1 mm precision), wing length using a ruler (±1 mm), and body mass using a hanging Pesola scale (±1 g). We collected blood (25 µl) from the basilic vein using a small gauge (27.5) needle to puncture the vein before drawing the blood into a capillary tube. Blood was preserved in 95% ethanol (1.5 ml). Finally, we took photographs of the general appearance of the bird.

Control individuals

Red phalaropes were opportunistically collected after being found dead during the breeding season at Barrow, Alaska in 2011 (male) and 2012 (female). We confirmed the sex of those control carcasses by visual inspection of their reproductive systems. Samples of muscles were collected during dissection, preserved in tissue preservation buffer (240.24 g Urea, 100 ml 1M Tris HCl pH 8.0, 11.69 g NaCl, 3.72 g EDTA, 5 g N-Lauroyl-sarcosine, npH2 O to 1 Liter). We used these samples as known-sex positive controls for PCR-based sex determination of the ambiguous individual.

PCR-based sex determination

Three birds were sexed using PCR-based methods: one control male, one control female, and the ambiguous bird. A small piece of tissue (close to 1 mm3) of the control birds was washed with 50 µl sterile water and centrifuged for 3 min at 10,000 rpm. Water was removed and the tissue washed a second time to remove any remaining salts from the preservation buffer that could have interfered with the PCR reaction. The tissue was then broken down using the point of a sterile tip in 50 µl of sterile water. Blood samples of the ambiguous bird were properly mixed and 50 µl were transferred to a new tube and centrifuged for 3 min at 10,000 rpm. Ethanol was removed and pelleted red blood cells were re-suspended in 50 µl DEPC water. The mixtures produced for each bird were incubated for 20 min at 55 °C with constant shaking and 5 µl was directly used as DNA template for the PCR reactions.

Sex determination was carried out according to Fridolfsson & Ellegren (1999), with minor modifications. 25 µl reactions contained 12.5 µl Amresco Hot Start Taq Master Mix 2x (Amresco LLC., Solon, Ohio, USA), 0.5 µM of each primer and 5 µl of the DNA template or of sterile water (negative control). Sequences of the primers used were 2550F:5′-GTTACTGATTCGTCTACGAGA-3′ and 2718R:5′-ATTGAAATGATCCAGTGCTTG-3′. PCR conditions were as follows: 94 °C for 1 min of initial denaturation, 35 cycles at 94 °C for 30 s, 50 °C for 30 s and 72 °C for 1 min, followed by a final extension at 72 °C for 5 min. PCR products were finally separated using 1.2% agarose gel electrophoresis with GelRed™ nucleic acid stain (Biotium, Inc., Hayward, California, USA). The sex of the ambiguous bird was determined by comparing PCR products against those amplified from the pattern displayed by the control male and female.

Permits

The Université de Moncton Animal Care Committee (permit #14-05) and Environment Canada (Scientific permit to capture and band migratory birds, #10872) approved capture techniques and immobilization procedures. We carried out red phalarope collections in Alaska under federal and state permits issued to R. Lanctot. The Department of Environment—Government of Nunavut (permit #WL-2014-039) and the Canadian Wildlife Service (permits #NUN-SCI-14-04) approved field research.

Results

Nest density

In summer 2014, the density of red phalarope nests on our study plots averaged 25 nests/km2 (SD = 11, n = 12 and 4 nests in the 36 and 24-ha plots, respectively). The ambiguous individual incubated in a nest located approximately 0.6 km outside both nest plots.

Nesting behavior

The ambiguous individual (Fig. 1) incubated four eggs in a nest found at the beginning of its incubation by flushing the bird on 27 June 2014. We revisited the nest on 2 July, 7 July, and 12 July when we saw signs of hatching of the eggs. We then visited the nest every 1–2 days until hatching on 17 July. We observed the brightly colored individual incubating its nest at every visit except on 13 July when the nest was not being attended.

Physical characteristics

The ambiguous individual was observed walking with an apparent handicap and upon capture, we noted that two digits of its right foot were missing second phalanges. Its bill and wing lengths overlapped with values reported for male and female red phalaropes trapped in Igloolik in a prior study (J Dale in Tracy, Schamel & Dale, 2002), while body mass was on average 7.9 g and 12.2 g lower than those of males and females, respectively (Table 1). Plumage patterns indicated the ambiguous bird had the red body feathers and plain black crown indicative of a female, but wing feathers resembling male feathers (Fig. 1).

Table 1 Morphometric measurements (±SD) of red phalaropes captured in Igloolik in a previous study (Tracy, Schamel & Dale, 2002) compared to those of the ambiguous individual measured in 2014.

Sample sizes are within brackets.

	Previous study	Ambiguous individual	
Bill length (mm)			
Male	22.2 ± 1.5 (48)	23.5	
Female	22.7 ± 1.2 (14)	
Wing length (mm)			
Male	128.4 ± 2.3 (48)	130	
Female	134.9 ± 2.9 (14)	
Body mass (g)			
Male	52.9 ± 3.8 (45)	45	
Female	57.2 ± 4.7 (13)	

PCR-based sex determination

The male and female positive controls exhibited the expected discriminative band pattern. The W-chromosome product amplified in the female was smaller than the Z-chromosome fragment amplified in the male, with sizes of approximately 300 bp and 550 bp, respectively (Fig. 2). The PCR product amplified from the ambiguous bird was identical to that of the control male, namely with a single product of around 550 bp (Fig. 2). No amplicon was observed in the negative control (Fig. 2).

Figure 2 PCR sex determination for red phalaropes at Igloolik, Nunavut, Canada and Barrow, Alaska.

PCR products were separated with a garose gel electrophoresis and stained with GelRed™ nucleic acid (Biotium, Inc., Hayward, California, US; see ‘Methods’) using sexing primers specific to birds (2550F/2718R; Fridolfsson & Ellegren, 1999). M: molecular marker, 1: typical male sampled in Barrow (550 bp), 2: the ambiguous bird (550 bp), 3: typical female sampled in Barrow (300 bp), and 4: negative control.

Discussion

According to PCR-based sexing, the brightly colored, ambiguous red phalarope was a male. Our result, combined with previous observations of brightly colored red phalarope males (Forbes et al., 1992; Tracy, Schamel & Dale, 2002), stresses the need for conducting more systematic genetic sexing combined with a standardized description of red phalarope plumage. This is of particular importance as the characteristic that is considered diagnostic of even bright males, namely the mottled crown, was not observed in the male described in this study.

There are a variety of reasons why bright male plumage might occur in this species. Johns (1964) showed that an injection of testosterone could experimentally induce the red nuptial feathers in phalaropes. Whether the bright plumage observed in the male red phalarope in our study could be associated with testosterone remains to be determined. In addition, it is unknown whether the physiological mechanism (testosterone or another mechanism) behind this feather coloration is a result of phenotypic plasticity, genetic variation or both. Yet, observations of eight males, including males from Igloolik Island (J Dale, pers. comm., 2016), whose distinctive plumage coloration was maintained over successive breeding seasons suggest that plumage coloration is genetically determined (Tracy, Schamel & Dale, 2002). It is interesting that our ambiguous male had cryptic wing feathers like his male counterparts (Fig. 1); such wing feather coloration would provide the necessary camouflage to avoid predation while incubating a nest. Further studies are required to sex individuals displaying such wing coloration and other potential distinguishing criteria between males and females currently discussed among shorebird biologists but as yet unpublished (e.g., tawny stripes on the back). This is especially needed on Igloolik Island as occasional observations conducted during summer 2015 in this location point to the possibility that male feather coloration is highly variable (Lecomte & Giroux, 2015, unpublished data), suggesting that our ambiguous male is not a singularity.

Redder males are thought to be of higher quality in species characterized by typical sex roles such as the bar-tailed godwit Limosalapponica (Piersma & Jukema, 1993). Yet, determining whether a female-like coloration would be associated with any variations in reproductive traits for males remains to be studied in the red phalarope (see an equivalent study in ruffs Philomachus pugnax: Küpper et al., 2016). To better understand the mechanisms inducing bright feather coloration in males, further studies are needed to compare physiological parameters and hormonal levels in bright individuals compared to typical bright females and dull males, and mate selection and breeding success of variously patterned males.

The PCR method used to sex these three individuals is based on the detection of a difference of intron size in similar copies of a gene found on the W and Z sex chromosomes (CHD1W and CHD1Z, respectively). This method proved to be successful for sex discrimination of most of the non-ratite bird species assayed by Fridolfsson & Ellengren (1999). When successful amplification occurs, a single PCR product is amplified in males, characteristic of their ZZ sexual chromosomes. For females, that have ZW sex chromosomes, two PCR products are usually amplified, with the largest product corresponding to the Z-chromosome, as in males. Females can sometimes display a single band pattern, when the W chromosome is preferentially amplified over the Z chromosome. In these cases, the single female product band is smaller than the single male product, still allowing the robust discrimination of both sexes (Fridolfsson & Ellegren, 1999). Dawson et al. (2001), however, reported some exceptions in which this method did not result in amplification products of different sizes, highlighting the importance of using positive controls of both sexes to validate the assay. In our study, the pattern exhibited by the control female was not unexpected and allowed differentiation of males from females. We repeated DNA amplification of the ambiguous individual using the same set of primers, and it resulted in the same 550 bp band (data not shown). We are thus confident of the sexual assignment of our ambiguous bird as a male.

Our result indicates that in some situations plumage dichromatism alone may not be sufficient to distinguish red phalarope males from females. Identifying diagnostic plumage characteristics of males would require range-wide studies scoring plumage of genetically sexed individuals with standardized protocols (Reynolds, 1987; Troscianko & Stevens, 2015). We also recommend further work to determine whether such female-like plumage on males are a result of phenotypic plasticity or genetic variation, and whether brightly colored males derive reproductive benefits from their coloration.

We thank M-C Frenette and M Trottier-Paquet for their valuable assistance in the field, the Government of Nunavut for their logistical support, D Edwards for sharing his thoughts about our observation, as well as four anonymous reviewers, W Goymann, J Dale, and G Rebstock that commented on a previous version of the manuscript. K Sage and S Talbot from the US Geological Service, Alaska Science Center provided samples from archived red phalaropes. The findings and conclusions in this article are those of the authors and do not necessarily represent the views of the US Fish and Wildlife Service.

Additional Information and Declarations

Competing Interests

Author Contributions

Animal Ethics

Field Study Permissions

Data Availability

The authors declare there are no competing interests.

Marie-Andrée Giroux conceived and designed the experiments, performed the experiments, analyzed the data, wrote the paper, reviewed drafts of the paper.

Delphine Ditlecadet performed the experiments, analyzed the data, wrote the paper, prepared figures and/or tables, reviewed drafts of the paper.

Luc J. Martin contributed reagents/materials/analysis tools, wrote the paper, reviewed drafts of the paper.

Richard B. Lanctot wrote the paper, reviewed drafts of the paper.

Nicolas Lecomte conceived and designed the experiments, performed the experiments, analyzed the data, contributed reagents/materials/analysis tools, wrote the paper, prepared figures and/or tables, reviewed drafts of the paper.

The following information was supplied relating to ethical approvals (i.e., approving body and any reference numbers):

Capture techniques and immobilization procedures were approved by the Université de Moncton Animal Care Committee (permit # 14-05) and by Environment Canada (Scientific permit to capture and band migratory birds, #10872). Red phalarope collections in Alaska were done under federal and state permits issued to R Lanctot.

The following information was supplied relating to field study approvals (i.e., approving body and any reference numbers):

Field research was approved by the Department of Environment – Government of Nunavut (permit # WL-2014-039) and the Canadian Wildlife Service (permits #NUN-SCI-14-04).

The following information was supplied regarding data availability:

Data are all directly included in the manuscript (see Results and Figures).

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
