# Peer review of "Sexing a sex-role-reversed species based on plumage: potential challenges in the red phalarope"

_PeerJ, doi:10.7717/peerj.1989_

## Round 0.1 · original submission · Minor Revisions

Please address all the reviewers' comments as they will improve the paper. Reviewer 3 kindly corrected the grammar and wording in a number of places (see attached annotated pdf), but there are additional typos or grammar errors. In particular, pay attention to agreement in number between subject and verb (singular/plural). Reviewer 3 questioned why the individual was considered ambiguous on lines 66-67 if both plumage and behavior were typical of females. I wonder if you meant the plumage and crown typical of a female but the "incubation behavior typical of a male".

·

Basic reporting

This manuscript describes a curious observation regarding a suspected female-like plumage in male red phalaropes. The authors describe a case of an incubating “ambiguous” phalarope that displays female like plumage characteristics but behaves like a typical male in this sex-role reversed species. By sampling this individual and comparing its genetic sexing profile with a typical male and a typical female phalarope the authors proof that the “ambiguous” individual indeed is a male.
The manuscript is well written, the introduction gives relevant background, the methods are very clear. The results are supported by relevant figures and graphs of high quality that also represent the raw data. Whether the manuscript otherwise conforms to the structure required by PeerJ, should be evaluated by an editor. In my view this is a very straight-forward manuscript that describes a very curious observation, shows that it is actually true that male phalaropes can look like females and that future research in this highly interesting species should be very cautious when relying on plumage characteristics for sexing the animals.

Experimental design

The study represents original primary research within the scope of PeerJ. The question is well designed, relevant and meaningful and fill an identified knowledge gap. As far as I can judge the investigation was performed to a high technical and ethical standard, and is described with sufficient detail.

Validity of the findings

The dara are robust and sound, statistical analyses would not make sense for such kind of data. The conclusions are well stated and linked to the original research question and results. There is no speculation.

Additional comments

Title: The term “gender” refers to a concept from social and/or psychological sciences and is specifically human. You cannot ask an animal about its “gender”. In animals we can only distinguish two sexes, hence the title should read “Sexing a sex-role-reversed species based on plumage: potential challenges in the red phalarope”

Introduction: The citations regarding sex roles in the first paragraph are somewhat arbitrary, especially Eens and Pinxten 2002, which is more about physiology. Would be better to cite the original studies that framed the concept (e.g. to be found in Andersson’s Sexual selection book) or reviews of the topic, such as the cited Kvarnemo and Ahnesjö paper.

Results:
Line 144: how could the nest be unattended on June 13th, when it was only found on June 27th? Do you mean July 13th?

Table 1: please indicate what the numbers refer to, i.e. millimeters and grams, I assume

·

Basic reporting

The article was well written and the English was very clear. A couple of typos, but otherwise fine.

Experimental design

Fine.

Validity of the findings

Fine - conclusions are clear and appropriately stated.

Additional comments

This is a nice study using molecular sexing to clearly confirm that males can be as brightly colored as females in red phalaropes. This is important because there are questionable reports of female incubation in this species - but these observations were more likely a result of misidentifying brightly colored males as females.

One comment I have regarding the discussion is your mention of the possibility that testosterone plays a role in the expression of the red plumage. I think it is important to clarify here that testosterone probably plays a role in triggering the molt into nuptial plumage (hence, Johns (1964) demonstration of red feather growth after T injection). This experiment does not imply that variation in testosterone-levels during molt is related to variation in how red the plumage is. Indeed - as we point out in our Birds of North America monograph, a number of males that had distinctive plumage colors and that were seen over multiple breeding seasons looked very similar between years. This suggests that the highly variable breeding coloration in male red phalaropes is fixed and genetically determined.

Reviewer 3 ·

Basic reporting

The manuscript is well written and the writing is clear throughout. I have several suggestions on minor edits to the text and have included these in the attached pdf. One last minor point is that both American and British spellings appear in the text and will need to be standardised: American spelling is used e.g. for “coloration” on line 60 and “liter” on line 105, but British spelling is used elsewhere, e.g. “behaviour” on line 67.

Experimental design

The research question is well defined, though as noted in the PDF, I would advise highlighting at the end of the introduction (lines 66-73) the particular traits that made the bird of ambiguous sex appear “ambiguous” based on field observations.
The methodology is generally sound, though usually with molecular sexing, it is advisable to include negative as well as positive controls, and to confirm the results either by using more than one sexing marker or carrying out two repeat PCRs with the same marker.

Validity of the findings

Excepting the lack of a negative control and the small possibility of contamination from the male control sample, the findings appear valid.
The conclusion is well stated and the ultimate discussion point, that plumage observations can’t always be relied on and molecular sexing methods are a great way to more accurately determine sex, is an important message for researchers.

Additional comments

This is a clear and simple study illustrating an important point on the need for molecular methods to determine the sex of phalaropes, a species thought to exhibit sexual dimorphism strong enough to distinguish males from females. I recommend that this manuscript be accepted for publication in PeerJ following minor revisions.

Annotated reviews are not available for download in order to protect the identity of reviewers who chose to remain anonymous.

---

## Round 0.2 · Minor Revisions

You have addressed most of the reviewers’ comments satisfactorily and the manuscript is close to being ready to accept. Please make the following editorial changes to clean up the grammar and a few inconsistencies:

The title and abstract were changed in the manuscript, but it looks like you did not upload new copies on the submission web page. Please make sure they are consistent.
Line 23: Insert scientific name after “the red phalarope”.
Line 49: Insert scientific name after “The red phalarope”.
Line 58: Replace “between males” with “among males”.
Line 64: The Sutton 1932 citation is not in the reference list.
Line 85: Change to “within a 36-ha nest plot and a 24-ha nest plot”.
Line 91: Please write dates consistently as Day Month Year. I think you mean 16 July here, not 16 June.
Line 97: Insert “tube” after “capillary”.
Line 102: Change “reproductive system” to “reproductive systems”.
Line 103: Do you mean 100ml? (not 100mls?)
Line 116: The degree symbol did not convert correctly in the pdf.
Line 134: Are the Nunavut and Canadian Wildlife Service permit numbers reversed or is it coincidence that the Nunavut permit number starts with “WL” and the Wildlife Service permit number starts with “NUN”?
Line: 142: Change “4-egg” to “4 eggs”.
Lines 143-146: Write dates consistently.
Line 144: Change “hatching on the eggs” to “hatching of the eggs”.
Line 151: I think this citation should be “J. Dale in Tracy et al. 2002”. It’s not unpublished if it’s in Tracey et al.
Line 154: Delete “possibly” and change to “wing feathers resembling male feathers”.
Line 166: Insert “a”: “combined with a standardized description”.
Line 175: Delete “possibly”.
Line 179: Change to “shorebird biologists”.
Line 184: Delete “a”: “in species characterized”.
Line 188: Insert scientific name of ruff.
Line 191: Change to “variously patterned males”.
Line 193: Change to “intron size”.
Line 194: Change “gender” to “sex”.
Line 211: Delete “for”.
Line 225: Place published?
Lines 234-236: Source (journal or book)?
Line 295: Change to “2=the ambiguous bird (550bp), 3=typical barrow female (300bp),…”.

I’m not satisfied with how you addressed James Dale’s comment on the relationship between testosterone and feather color. Lines 171-172 still imply that testosterone level influences the brightness of the feathers, which was not included in Johns’ 1964 study. Testosterone levels in that study were constant so it did not show that higher testosterone levels resulted in brighter plumage. If you want to speculate that higher testosterone levels result in brighter plumage, it should be reworded to be clear that it’s speculation. Don’t use the word “hence”, which implies that it’s a conclusion from the Johns study.

·

Basic reporting

The paper is much improved over the previous version.

Experimental design

Fine.

Validity of the findings

Fine.

Additional comments

I think this is a nice study and the authors should be commended for taking on the various comments from the reviewers and improving the paper considerably.

I do however suggest that you revisit your decision not to include the idea that the highly variable breeding coloration in male red phalaropes may be fixed and genetically determined. As you correctly point out, Ridley (1980) indicated that male variants were useful for individual recognition within a single year. However if you look at the end of that same paragraph in the BNA monograph we state: "Variations maintained in 8 distinctly plumaged males over successive breeding seasons (JD, DS and DMT), suggesting plumage color and pattern is fixed and genetically determined." This is based on our own observations, and included a number of males from the same population you studied.

Reviewer 3 ·

Basic reporting

The manuscript remains well written and clear. Just a few very minor typos are still present (including line 209 "characteristics for of males" and a missing space after "2550F:" on line 119).

Experimental design

The authors have addressed previous comments thoroughly, making updates to the manuscript on the inclusion of a negative control and their use of repeated sex-typing with the same marker to confirm sex assignment for the ambiguous bird. The methods are well described and technically sound.

Validity of the findings

Following updates by the authors, the findings do appear robust and statistically sound.

Additional comments

The revised manuscript is now much improved, including clarifications in several areas. The authors have responded satisfactorily to all previous points. This is a nice, solid study highlighting the need for molecular methods in determining the sex of phalaropes.

---

## Round 0.3 · Minor Revisions

This manuscript is very close to being acceptable. I’m being picky about it because PeerJ does not do copyediting. Please make the following minor edits:

Abstract: You added the scientific name of the red phalarope to the abstract, but did not upload a new abstract file. Please upload it.
Line 119: Please indent new paragraph.
Line 131: Delete space between # and 14-05.
Line 136: Delete space between # and WL-2014-039.
Line 141: Insert “in”: “incubated in a nest”.
Line 143: Change to: “incubated four eggs in a nest found …”.
Line 154: Insert “the”: “… bird had the red body feathers”.
Lines 175-177: Spell out eight. The Tracy et al. account in Birds of North America does not say the eight males were on Igloolik Island. Do you know that that’s were they were?
Acknowledgments: You should add the new reviewers.

---

## Round 0.4 · accepted · Accept

Thanks for your attention to the details.